# Efficacy of Intact Cord Resuscitation Compared to Immediate Cord Clamping on Cardiorespiratory Adaptation at Birth in Infants with Isolated Congenital Diaphragmatic Hernia (CHIC)

**DOI:** 10.3390/children8050339

**Published:** 2021-04-26

**Authors:** Kévin Le Duc, Sébastien Mur, Thameur Rakza, Mohamed Riadh Boukhris, Céline Rousset, Pascal Vaast, Nathalie Westlynk, Estelle Aubry, Dyuti Sharma, Laurent Storme

**Affiliations:** 1ULR2694 Metrics‑Perinatal Environment and Health, University of Lille, 59000 Lille, France; estelle.aubry@chru-lille.fr (E.A.); dyuti.sharma@chru-lille.fr (D.S.); laurent.storme@chru-lille.fr (L.S.); 2Department of Neonatology, Jeanne de Flandre Hospital, Centre Hospitalier Universitaire de Lille, 59000 Lille, France; sebastien.mur@chru-lille.fr (S.M.); Thameur.RAKZA@chru-lille.fr (T.R.); riadh.boukhris@chru-lille.fr (M.R.B.); Celine.rousset@chru-lille.fr (C.R.); 3Center for Rare Disease Congenital Diaphragmatic Hernia, Jeanne de Flandre Hospital, Centre Hospitalier Universitaire de Lille, 59000 Lille, France; pascal.vaast@chru-lille.fr; 4Department of Obstetrics, Jeanne de Flandre Hospital, Centre Hospitalier Universitaire de Lille, 59000 Lille, France; nathalie.westlynk@chru-lille.fr; 5Department of Pediatric Surgery, Jeanne de Flandre Hospital, Centre Hospitalier Universitaire de Lille, 59000 Lille, France

**Keywords:** intact cord resuscitation, delivery room resuscitation, congenital diaphragmatic hernia

## Abstract

Resuscitation at birth of infants with Congenital Diaphragmatic Hernia (CDH) remains highly challenging because of severe failure of cardiorespiratory adaptation at birth. Usually, the umbilical cord is clamped immediately after birth. Delaying cord clamping while the resuscitation maneuvers are started may: (1) facilitate blood transfer from placenta to baby to augment circulatory blood volume; (2) avoid loss of venous return and decrease in left ventricle filling caused by immediate cord clamping; (3) prevent initial hypoxemia because of sustained uteroplacental gas exchange after birth when the cord is intact. The aim of this trial is to evaluate the efficacy of intact cord resuscitation compared to immediate cord clamping on cardiorespiratory adaptation at birth in infants with isolated CDH. The Congenital Hernia Intact Cord (CHIC) trial is a prospective multicenter open-label randomized controlled trial in two balanced parallel groups. Participants are randomized either immediate cord clamping (the cord will be clamped within the first 15 s after birth) or to intact cord resuscitation group (umbilical cord will be kept intact during the first part of the resuscitation). The primary end-point is the number of infants with APGAR score <4 at 1 min or <7 at 5 min. One hundred eighty participants are expected for this trial. To our knowledge, CHIC is the first study randomized controlled trial evaluating intact cord resuscitation on newborn infant with congenital diaphragmatic hernia. Better cardiorespiratory adaptation is expected when the resuscitation maneuvers are started while the cord is still connected to the placenta.

## 1. Introduction

Congenital Diaphragmatic Hernia (CDH) is a rare disease (1/3000 pregnancies) caused by diaphragmatic defect with ascension of the abdominal content into the thoracic cavity. Pulmonary consequences of CDH present a broad spectrum of severity [1]. Despite major improvement in neonatal intensive care, mortality and morbidity related to failure of transition at birth remain high with a mortality rate between 20 to 40% [2]. Failure of transition at birth results from persistent pulmonary hypertension (PPHN) and is almost a universal finding in CDH [3,4]. There is an urgent need for additional research in order to promote cardiorespiratory adaptation at birth.

The CDH EURO Consortium proposes intubating newborn infants with CDH immediately after birth to limit PPHN [5]. Treatment in the delivery room is directed at reaching an adequate oxygenation while avoiding high airway pressures. Low peak pressures are given to avoid lung damage to the hypoplastic and contralateral lung [4]. Resuscitation at birth remains highly challenging because of severe failure of cardiorespiratory adaptation at birth. Despite the recommendations, aggressive resuscitation if often applied because the baby is frequently cyanotic and bradycardic as soon as the umbilical cord is sectioned. Pneumothoraces—a marker of barotrauma—can occur early after starting the resuscitation maneuvers and is associated with high mortality rate [6]. A postmortem CDH study has shown that the high mortality in CDH can be partially attributed to pulmonary barotrauma causing damage to hypoplastic lungs [7]. Therefore, the immediate postnatal period of resuscitation represents a window of extreme vulnerability of the baby with CDH, conditioning the short- and long-term outcome.

Traditionally, the umbilical cord is clamped and cut immediately after birth. Clamping the umbilical cord immediately increases systemic peripheral resistance, resulting in an increase in arterial pressure (afterload). However, as the placental circulation receives 30–50% of fetal cardiac output, cord clamping transiently reduces venous return (by 30–50%), which, combined with the increase in afterload, decreases cardiac output [8]. Following cord clamping, umbilical venous return is lost and left ventricular output becomes dependent on pulmonary blood flow, as in adults. Any delay between umbilical cord clamping and the increase in pulmonary blood flow could therefore severely affect left ventricular output and potentially result in organ injury. In CDH infant, increase in pulmonary blood flow is delayed after birth because of PPHN. These changes may significantly impact on cardiac function after clamping of the cord.

In 2015, the International Liaison Committee on Resuscitation (ILCOR) recommended that the cord should not be cut for at least 1 to 3 min after birth in infants not requiring resuscitation [9]. This recommended change in practice is to facilitate blood transfer from placenta to baby to reduce iron deficiency and later anemia in the full-term newborn infant. In the same way, meta-analysis indicate that delayed cord clamping in the preterm infant increases circulating volume and improves blood pressure, reduces the need for blood transfusion, risk of intraventricular hemorrhage and necrotizing enterocolitis [10,11].

The ex-utero intrapartum treatment (EXIT) procedure has been proposed during a variety of surgical procedures, mainly cervical mass resection, performed at birth to secure the fetal airway or ensure successful transition to postnatal environment [12]. The aim of the EXIT procedure is to maintain placental gas exchange while steps are taken to optimize the transition of the newborn infant from fetal to neonatal life.

### Preliminary Data Obtained

We have shown in three previous studies in normal newborn lambs and in newborn lambs with PPHN that blood gases did not change 30 min after birth despite lack of breathing compared to the blood gases obtained before birth [13,14,15].

We also evaluated the safety, feasibility and impacts of intact cord resuscitation (ICR) on cardiorespiratory adaptation at birth in newborn infants with CDH in a prospective pilot study [16]. 

Resuscitation before cord clamping was possible for all infants in the ICR group. The cord was clamped at 7 ± 3 min after birth. No increase in maternal or neonatal adverse events was observed during the period of ICR. 

A multicenter randomized clinical study is required to confirm the benefit of intact cord resuscitation in CDH infants on cardiorespiratory adaptation at birth. To know whether intact cord resuscitation improves initial cardiorespiratory adaptation at birth is a major issue. 

## 2. Materials and Methods

### 2.1. Objectives

#### 2.1.1. Primary Endpoints

The primary endpoint is the rate of infants with APGAR score < 4 at 1 min or <7 at 5 min.

This primary objective is to show the efficacy of intact cord resuscitation compared to immediate cord clamping on cardiorespiratory adaptation at birth in full term newborn infants with isolated CDH. APGAR score is used to assess cardiorespiratory adaptation at birth. APGAR score is based on clinical assessment of color, heart rate, grimace, muscle tone, and respiratory effort. It is used worldwide by neonatal caregivers, both as a measure of the infant’s clinical status as well as a measure of the infant’s response to resuscitation. APGAR score can be assessed during resuscitation maneuvers, including intubation. Slight inter-observer variation may exist in intubated baby mainly due to lack of standardization of scoring respiratory effort states. In order to limit inter-observer variability, we will standardize respiratory effort scoring as previously reported [17]. In order to optimize APGAR scoring, each of its five items will be recorded by two observers. A timer is switched on at birth. Apgar will be assessed at 1, 5 and 10 min. 

APGAR is universally recognized as a major prognostic variable of neonatal outcome [18]. More specifically, APGAR scoring remains the main early predictor for mortality in isolated CDH infant [19]. Low APGAR score (<7 at 5 min) is associated with an increased risk of death (OR 2.7 (1.9–4)) in a cohort of 2202 infants with CDH.

#### 2.1.2. Secondary Endpoints

The secondary objectives of the studies are as follows:To ensure maternal safety of the procedure, blood loss will be carefully monitored after birth. A graduated collector bag for blood, placed under the woman’s buttocks just after delivery of the child, will be used systematically to measure the blood lost through the vagina in the immediate postpartum period. This bag will be left in place at least for 15 min. The following maternal safety endpoints will be assessed:
-Frequency of postpartum hemorrhage (PPH) defined by blood loss ≥ 500 mL;-Frequency of severe PPH, defined by measured blood loss ≥ 1000 mL;-Blood loss volume at 15 min after birth;-Total postpartum blood loss volume (at bag removal);
To assess the effect of intact cord resuscitation compared to immediate cord clamping on cardiorespiratory adaptation of infants after birth, the following secondary endpoints will be assessed:
-Frequency of infants with the need for epinephrine administration and/or fluid resuscitation;-Frequency of infants with the need for chest compressions;-Pre-ductal SpO2, and heart rate at 1, 5, and 10 min after birth: a pulse oxymeter sensor will be placed at the right hand as soon as possible (within the first minute after birth), which then will be connected to a pulse oxymeter; -Blood gases and plasma lactate concentration at one hour after birth (H1): these quantitative variables can be considered as objective markers of early cardiorespiratory adaptation at birth. -Blood gases, pre- and postductal SpO2, lactate, FiO2 set to obtain preductal SpO2 90–94%, ventilatory parameters (peak inspiratory pressure, respiratory rate), heart rate, blood pressure, and urine output at H1, H24, H48, H72, D7, D28;-Volume of fluid resuscitation during the first 24 h;-Frequency of infants with the need for vasoactive drugs during the first 24 h;-Frequency of infants with the need for pulmonary vasodilator during the first 24 h;-Hemoglobin concentration at H24;-Echocardiographic parameters (left and right mean blood flow velocities, pulmonary artery pressure) at H6, H24, H48, D7, D28;
To assess the effect of intact cord resuscitation compared to immediate cord clamping on infants’ mortality and morbidity, we choose to assess the number of free-days from medical support in order to address the effect of mortality on the assessment of morbidity. Because of the randomization process, we can assume that the two groups will be similar in term of severity. In the hypothesis that intact cord resuscitation decreases the mortality, it means that more severe CDH infant may survive, which may in turn increase the apparent morbidity. In case of death, the number of free-days from medical support is zero, which will not modify the number of free-days from medical support in the surviving population. Therefore, the following secondary end-points will be assessed:
-Infant mortality rate at 90-day after birth;-Infant morbidity outcomes assessed within the first 90 days after birth:
○mechanical ventilation free-days (defined as days alive and free of mechanical ventilation from birth to 90 days), ○extracorporeal membranous oxygenation free-days (defined as days alive and free of extracorporeal membranous oxygenation from birth to 90 days)○pulmonary vasodilator treatment free-days (defined as days alive and free of pulmonary vasodilator treatment including inhaled NO, sildenafil, prostacyclin analog, bosentan from birth to 90 days),○O2 supplementation free-days (defined as days alive and free of O2 supplementation including non-invasive respiratory support from birth to 90 days),○parenteral nutrition free-days (defined as days alive and free of parenteral nutrition from birth to 90 days) ○Total duration of hospitalization,
To assess parental acceptability and psychological impact of starting resuscitation while the cord is intact as compared to the immediate cord clamping group:
-Number of refusals to participating with the protocol: the reasons for refusal will be recorded (do not want to participate to a research protocol, to not want to be randomized in the immediate cord clamping group, to not want to be randomized in the intact cord resuscitation group); -Anxiety and depression level assessed by Hospital Anxiety and Depression Scale, HADS. HADS is an auto-questionnaire translated in French, including seven questions to assess anxiety and seven questions to assess depression. HADS questionnaire lasts 3 to 6 min, and is mostly well accepted. Both parents will be requested to answer the questionnaire within the first 3 days after birth, in a calm and neutral room at the maternity ward. Psychologist in charge of the parents will be informed of the results of the test to adapt family support if required; -Semi-structured interviews will be proposed by a psychologist to the parents whatever the issue, at the end of the study period (90 ± 7 days after birth) to assess their personal experience of the resuscitating period at birth, including both early/delayed cord clamping and close/remote resuscitation maneuvers. The interviews tape recordings will be transcribed and then analyzed using thematic coding. The anonymized data will be independently coded by three researchers and compared for consistency of interpretation. The themes that emerged following the final coding will be used for a qualitative analysis of the parental verbatim;


### 2.2. Trial Design

This is a prospective multicenter open-label randomized controlled trial in two balanced parallel groups. 

### 2.3. Inclusion Criteria

Antenatal diagnosis of CDHNo severe additional malformation or chromosomal diseasesFull term (>36 weeks gestational age)No inclusion in another antenatal trialWritten informed consents from the parentsThe fetuses which required Fetoscopic Endoluminal Tracheal Occlusion (FETO procedure) will also be included in the study.

### 2.4. Exclusion Criteria

Preterm birth before 37 weeks gestational ageOther severe malformation(s) or chromosomal diseasesNeed for emergency cesarean section (“red code”)Twin pregnancyPostnatal diagnosis of additional severe malformation(s) or chromosomal diseases

### 2.5. Study Organization

After assessing the inclusion and exclusion criteria, the parents will be informed of the study protocol and the intact cord resuscitation procedure (between 22- and 36-weeks gestational age). The written parental consents are requested before 37 weeks gestational age. A web-based randomization with electronic case-report form (eCRF) system is performed at admission for delivery at the maternity ward. Prenatal o/e LHR on ultrasounds and lung volumetry on fetal MRI allow to classify patients in according the severity of the CDH (mild, moderate, severe and extremely severe) [20,21]. Randomization is stratified according to center, and four antenatal severity subgroups defined by ultrasound lung-to-head ratio Observed/Expected and liver position (Extreme <15%, Severe 15–25%, Moderate 25–45%, and Mild >45%). 

The infants with CDH will be included in the study and randomized in one of the two groups. Severity subgroups is considered as a stratified factor in randomization. A dynamic randomization procedure using the Pocock and Simon minimization method is used [22]. Whatever the group of randomization, the national guidelines [23,24] for resuscitation maneuvers in the delivery room will be applied similarly in both groups. Except the timing of cord clamping, the overall management including the resuscitation maneuvers in the delivery room (intact cord resuscitation group) or resuscitation room (early cord clamping group), and organization of the follow-up will be similar in both groups. In particular, the main resuscitation maneuvers at birth are:The newborn infant should be intubated routinely without bag and mask ventilation, with an endotracheal tube 3.5 mm;Ventilation in the delivery room should be done with a peak pressure below 25 cmH2O, and an initial FiO2 = 1;The goal of treatment in the delivery room is achieving heart rate > 120 beats/min and increasing preductal SpO2 or achieving acceptable preductal SpO2 targets between 80 and 95%;An oro- or nasogastric tube with continuous or intermittent suction should be placed;
Group 1: Immediate cord clamping with transfer to the resuscitation room

In the immediate cord clamping group, the cord will be clamped within the first 20 s after birth and the infant will be transferred to the resuscitation room. The newborn infant will be intubated and mechanically ventilated as quickly as possible on the resuscitation table as recommended in the Programme National de Soins. After cardiorespiratory stabilization (heart rate > 120/min, increasing preductal O2 saturation or achieving acceptable preductal SpO2 targets between 80 and 95%), the infant will be transferred to the neonatal intensive care unit (NICU). Oxytocin is infused to the mother as recommended in the local protocol (usually just after birth or cord clamping).

Group 2: Intact cord resuscitation with resuscitation maneuvers performed on a dedicated trolley placed close to the mother

In the intact cord resuscitation group, the umbilical cord will be kept intact during the initial phase of the resuscitation. The infant will be placed on a specifically designed compact trolley with a warmed platform, suitable for commencing resuscitation between the mother’s legs in case of vaginal birth or near the operating table beside the mother in case of cesarean section. This trolley will be fully equipped for resuscitation, including a suction device, gas flowmeter/blender, ventilator, and monitoring system. Its height can be adjusted to position the infant close to the maternal perineum. The infant will be intubated and mechanically ventilated on this trolley; special care will be taken to prevent stretching, compression, or kinking of the cord. The cord will be clamped after cardiorespiratory stabilization will be obtained (heart rate > 120/min, increasing preductal O2 saturation or achieving acceptable preductal SpO2 targets between 80 and 95%) or in case of spontaneous placental expulsion. Oxytocin is infused to the mother just after cord clamping. The infant will be then transferred to the NICU. See Figure 1 “Study organization”.

### 2.6. Interventions

Except the timing of cord clamping, the national guidelines will be applied during the course of the prenatal and postnatal management.

Some differences with usual care during implementation of the protocol are as follows:Here is a photograph of the trolley (LifeStart, Inspiration Healthcare, Great Britain) (Figure 2):

In both groups, although APGAR score is routinely assessed in every baby at 1 and 5 mn, special care will be taken to optimize scoring taken into account the fact that most infants with CDH are intubated at 5 min. APGAR score can be assessed during resuscitation maneuvers, including intubation. Slight inter-observer variation may exist in intubated baby mainly due to lack of standardization of scoring respiratory effort states [22]. In order to limit inter-observer variability, we will standardize respiratory effort scoring as previously reported [22]:an infant who is apneic and requires intubation and ventilation should receive the minimum value of 0 for respiratory effort;an infant who requires artificial ventilation at birth due to irregular or shallow ventilation should score 1;To assess whether an artificially ventilated infant is apnoeic or not, ventilation should be stopped briefly, when possible, to check for the presence of spontaneous respiratory movements (apneic, score = 0; irregular or shallow ventilation, score 1; spontaneous effective ventilation, score 2).Preductal SpO2 and heart rate are routinely assessed in every baby with CDH. They will be recorded at 1, 5 and 10 min after birth in both groups.Blood gases and lactate concentration are routinely assessed in every baby with CDH. These parameters will be recorded at H1, H12, H24, H48, H72, D7, D28, in both groups. Compared with usual care, no additional blood samples are required for the protocol;In both groups, although noninvasive echocardiography is routinely assessed in every baby with CDH, left and right mean blood flow velocities, pulmonary artery pressure will be recorded at H6, H24, H48, D7, D28. Compared with usual care, no additional echocardiography is required for the protocol;Acceptability by the parents and psychological impact of intact cord resuscitation will be assessed using a questionnaire and a semi-directive interview.

### 2.7. Criteria for Discontinuation of the Procedure

The healthcare providers (obstetrician, midwives, anesthetist, and pediatrician) in charge of the baby or the mother may stop the procedure—i.e., clamp the cord—if they judge it is required for the safety of the patient. The reason for stopping the procedure and the timing at cord clamping will be recorded. In that case, the patient is not excluded from the study and the variable is recorded as planned in the protocol. The patient data are analyzed according to intention to treat principle.

The follow-up of the infants is planned till the age of 3 months (90 days), corresponding to the last visit. If the patient is transferred in another hospital, the variable will be recorded till the age of 3 months.

### 2.8. Sample Size Calculation

The primary objective is to show the superiority of intact cord resuscitation (experimental group) compared to immediate cord clamping (control group) to improve cardiorespiratory adaptation at birth of CDH infant. The primary endpoint is defined as the rate of infants with low APGAR score, defined by a APGAR score < 4 at 1 min or <7 at 5 min of life. On the basis of literature, we expected that 37% of CDH infant in control arm will have a low APGAR score. To show a 50% relative reduction (corresponding to a rate of 18.5% in experimental group), with a type I error of 5% (two-sided test), and a power of 80%, we calculated that 90 subjects (mother/infant) per group are needed, which means a total of 180 subjects.

### 2.9. Statistical Analysis Plan

Statistical analyses will be independently performed by the Biostatistics Department of University of Lille under the responsibility of Professor Alain Duhamel. For the data analysis, statisticians will be unaware of the treatment group allocation. Data will be analyzed using the SAS software (SAS Institute Inc., Cary, NC, USA) and all statistical tests will be performed with a two-tailed alpha risk of 0.05. All analyses will be performed in all randomized patients based on their original group of randomizations, according to the intention-to-treat principle. No interim analysis will be performed. A detailed statistical analysis plan will be written and finalized prior to the database lock. 

Baseline characteristics will be described for each group. Quantitative variables will be expressed as mean (standard deviation), or median (interquartile range) for non-Gaussian distribution. Categorical variables will be expressed as frequencies and percentages. Normality of distribution will be assessed graphically and using the Shapiro–Wilk test.

### 2.10. Data Safety Monitoring Board

The Data and Safety Monitoring Board (DSMB) is an independent consultative board asked to express an opinion to the sponsor of the study on the benefit/risk ratio and the management of the clinical trial.

A DSMB will be set up, composed of the following members, at least: two pediatricians, two obstetricians, one anesthetist, and one methodologist.

Its members will not participate in the study and will be independent from the investigation centers. They will be nominated by the sponsor of the study and will participate as volunteers in the respect of confidentiality of the data. 

The DSMB will analyze the management of the study as well as its benefit/risk exposure and might recommend to stop the study in certain circumstances:-If the number of Serious Adverse Event (SAE) within the first 24 h after birth is twice higher in the “intact cord resuscitation” group after the inclusion of 30 patients,-If the DSMB judges that a reported SAE caused by the procedure requires to stop the study,-If inclusion rate is less than 25% of the total inclusion objectives, 24 months after starting the inclusion in the study. The board will meet after the inclusion of 30, 90 and 150 patients, to address those points, but it could be summoned on other circumstances:-If a mother’s death occurs within three months after she gave birth to her child,-At the request of any PI involved in the study.

## 3. Results

Presently, 11 centers are participating to the study: two additional centers are expected to include CDH infants. The first center started enrolling patients for 6 months. At the present time, 10 patients have been recruited. For now, parental consents have been obtained in all cases. Recruitment is expected to be completed by December 2023.

## 4. Discussion

Resuscitation at birth of infants with congenital diaphragmatic hernia remains highly challenging because of severe failure of cardiorespiratory adaptation at birth.

The traditional approach in the delivery room is immediate cord clamping followed by intubation. Initiating resuscitation prior to umbilical cord clamping (UCC) may support this transition, to avoid the loss of venous return and decrease in left ventricle filling caused by cord clamping, ideally increase in pulmonary blood flow should precede cord clamping. This would enable pulmonary venous return to replace umbilical venous return as the primary source for left ventricle preload and to minimize swings in left ventricular output caused by cord clamping. In infants with CDH, the cord is clamped while the pulmonary blood flow is still low. 

Evidence indicates that utero-placental gas exchange continues after birth when the cord is intact. A previous experimental study in newborn lambs showed that heart rates and right ventricle output were markedly decreased within 120 s of cord clamping. In contrast, if umbilical cord clamping was delayed until after ventilation had commenced, these large changes in heart rate, arterial pressures and flows were greatly reduced, resulting in a much more stable cardiovascular transition after birth [25,26]. We hypothesize that delayed cord clamping allows time for the infant to aerate its lungs and increase pulmonary blood flow before venous return from the placental circulation is lost.

Preliminary study has demonstrated the feasibility of intact cord resuscitation. Starting mechanical ventilation of the infant newborn with congenital diaphragmatic hernia before clamping the cord allowed a potential benefit to improved blood pressure at 1 h of life compared to those resuscitated traditionally [27].

Lefebvre et al. demonstrated that Intact cord resuscitation was associated with higher APGAR scores at 1 and 5 min after birth. The pH was higher and the plasma lactate concentration was significantly lower at one hour after birth in the intact cord resuscitation than in the immediate cord clamp group (pH = 7.17 ± 0.1 vs. 7.08 ± 0.2; lactate = 3.6 ± 2 vs. 6.6 ± 4 mmol/L, *p* < 0.05). Mean blood pressure was significantly higher in the intact cord resuscitation than in the immediate cord clamp group at H1 (52 ± 7 vs. 42 ± 7 mmHg), H6 (47 ± 3 vs. 40 ± 5 mmHg) and H12 (44 ± 2 vs. 39 ± 3 mmHg) (*p* < 0.05). Therefore, commencing resuscitation and initiating ventilation while the infant is still attached to the placenta is feasible in infants with CDH. The procedure is safe, and may support the cardiorespiratory transition at birth in infants with CDH [16].

To our knowledge, CHIC trial is the first prospective multicenter randomized controlled trial to assess the impact of delayed cord clamping during resuscitation of newborn infant with congenital diaphragmatic hernia. Future publication on the primary and secondary results of the CHIC trial study will make a substantial contribution to this important orientation of research and the results will provide clear evidence on the impact of delayed cord clamping during resuscitation of the newborn infant with congenital diaphragmatic hernia.

## Figures and Tables

**Figure 1 children-08-00339-f001:**
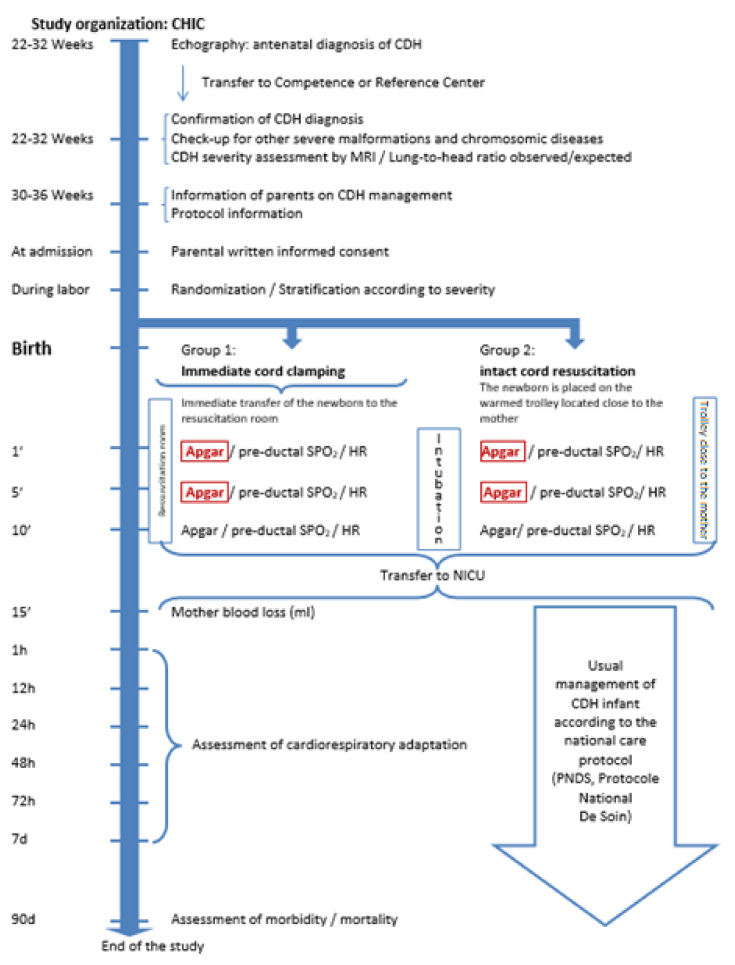
Study organization. Congenital Diaphragmatic Hernia (CDH); Congenital Hernia Intact Cord (CHIC); Blood oxygen saturation (SpO_2_); Heart Rate (HR); Neonatal Intensive Care Unit (NICU).

**Figure 2 children-08-00339-f002:**
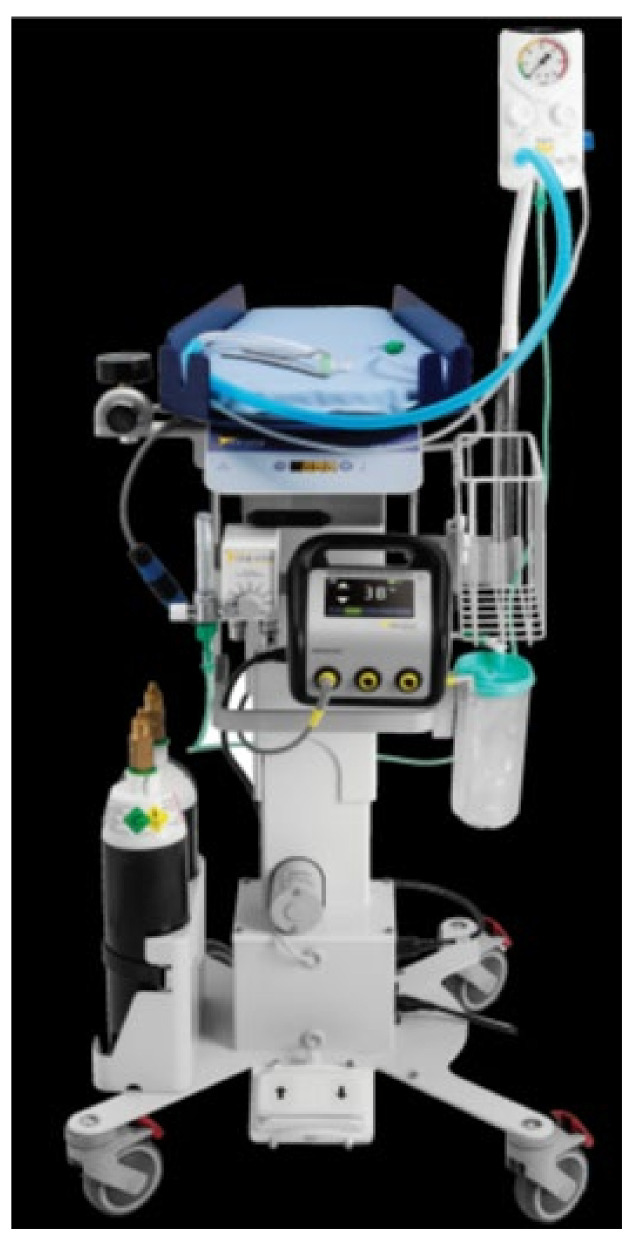
The resuscitation table Lifestart is easily maneuverable, with ergonomic top for placement of the baby. It is equipped with a neonatal warming system, a built-in timer that tracks clamping delay, and a suction device. The baby is warmed from under the patient, ensuring that there is no impediment to resuscitation maneuvers. The electrically operated raising and lowering mechanism allows the nursing platform to be easily positioned at the optimal height for each individual situation and adjusted as necessary. All the resuscitation equipment can be positioned close to the mother and adapted for all types of delivery: natural; assisted; or caesarean section.

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
