# Peer review of "Efficacy of Intact Cord Resuscitation Compared to Immediate Cord Clamping on Cardiorespiratory Adaptation at Birth in Infants with Isolated Congenital Diaphragmatic Hernia (CHIC)"

_children, 2021, doi:10.3390/children8050339_

Round 1

Reviewer 1 Report

The authors present a descriptive manuscript of a study protocol to be performed. It is a proposal for a prospective randomized control trial to assess the efficacy of intact cord resuscitation compared to immediate cord clamping on cardio-respiratory adaptation at birth in infants with isolated congenital diaphragmatic hernia. Being a protocol, no results are presented nor conclusions can be extracted.

The rationale and premises to develop this study are well exposed with a good introduction. The primary and secondary endpoints are clearly described. They divide the study in to arms, the intervention group (delayed cord clamping with resuscitation on placental support) versus the not-intervention group (just immediate cord clamping at delivery) independently of the mode of delivery by C-section or vaginal route.

The main concern is that there is a great variability in the severity of CDH depending on the degree of the pulmonary hypoplasia and pulmonary hypertension. To be appropriately balanced, and avoid any bias, both groups should have enough patients in each category of severity (mild, moderate, severe and extremely severe). These can be classified by prenatal o/e LHR on ultrasounds and volumetry on fetal MRI. They must guarantee that significant number of severe cases are included.

In Fig. 1, the authors wrongly included o/eLHR (in the prenatal evaluation) as Lung-to-HEART Ratio, when should be Lung to-to-HEAD Ratio. They state severity assessment based on MRI, but also should be based on prenatal u/s.

Other big concern that is not described or clarified in the manuscript, or in the inclusion/exclusion criteria, is if they will include CDH fetal patients that had prenatal intervention with tracheal balloon (FETO) in their study, or just simply they will exclude these ones.

They, with good judgement, excluded patients with associated malformations and less than 37 weeks' gestation at delivery to minimize confounding factors.

The sample size estimates a total of 180 patients. There is not described how much time the authors anticipate for completing this study and how many Centers they require (or already are participating) in the study. Actually, the recruitment starting date is determined for 6 months ago, Did the study already started, what is so far the rate of recruitment and parental acceptance? How many Centers involved? They plan to incorporate more?

The discussion and references seem appropriate. In summary, this study looks relevant and interesting, but should avoid bias for the variability of the spectrum of CDH in terms of severity, morbidity and mortality.

Reviewer 2 Report

Authors present the “CHIC”, that  is to our knowledge, the first study randomized controlled trial evaluating intact cord resuscitation on newborn infant with congenital diaphragmatic hernia.

It is important to point out that CHIC trial is a prospective multicenter open-label randomized controlled trial in two bal-anced parallel groups.

The background of this approach is based on the concepts that delaying cord clamping while the resuscitation maneuvers are started may: 1) facilitate blood transfer from placenta to baby to augment circulatory blood volume; 2) avoid the loss of venous return and decrease in left ventricle filling caused by immediate cord clamping; 3) prevent initial hypoxemia because of sustained uteroplacental gas exchange after birth when the cord is intact.

CONSIDERATIONS:

  • The Authors are part of a cutting-edge group and are experienced professionals in the subject;
  • The introduction of the paper, which contains the background, is complete and understandable;
  • Objectives of “ Matherials and Metods”, are complete. The authors could have organized the contents in the text in a more organic way, with a motivation for each single point;
  • Trial design: it seems clear to the reader;
  • I would have preferred to organize the paper with a "Results" section, rather than go from M&M to Discussion;

In conclusion, I think the paper is interesting and deals with an important topic, it needs fewer revisions to be published. The suggested changes mainly concern the structure of the paper, making the reading and interpretation of the results clearer.
